# Mediating Effect of Motivation on the Relationship of Fitness with Volitional High-Intensity Exercise in High-School Students

**DOI:** 10.3390/healthcare11060800

**Published:** 2023-03-08

**Authors:** André Bento, Luis Carrasco, Armando Raimundo

**Affiliations:** 1Comprehensive Health Research Centre (CHRC), Department of Sport and Health, School of Health and Human Development, University of Évora, 7004-516 Évora, Portugal; 2BIOFANEX Research Group, Department of Physical Education and Sport, University of Seville, Pirotecnia St., E-41013 Seville, Spain

**Keywords:** body composition, health, HIIT, physical education

## Abstract

We aimed to investigate the relationship between physical fitness and motivation in adolescents and analyze if the associations of physical fitness with volitional exercise intensity in adolescents are mediated by motivation. The participants were 108 adolescents (58 girls 16.0 ± 0.92 years). Cardiorespiratory fitness (CRF) was assessed using the Yo-YoITL1, and the push-up test was used to evaluate strength. Body composition was measured by bioelectrical impedance analysis. The intervention was applied in the first 10–15 min of each Physical Education class (PEC), twice a week, for 16 weeks and ranged from 14 to 20 all-out bouts intervals, adopting a 2:1 work to rest ratio. A cut-point of ≥90% of the maximal heart rate (HR) was used as a criterion for satisfactory compliance with high-intensity exercise. Volition intensity was assessed through a forearm wearable plethysmography heart rate sensor to ensure compliance with the exercise stimulus at the predetermined target HR zone. Motivation was estimated with a validated questionnaire (BREQ-3). Mediation effects were estimated using bootstrapped 95% confidence intervals and were deemed significant if zero was not included in the intervals, and values below 0.05 were considered to indicate statistical significance. The mediation analysis revealed a non-significant indirect effect of physical fitness through motivation on exercise intensity, specifically on CRF (B = −0.0355, 95% BootCI [−0.5838; 0.4559]), muscular fitness (B = −0.7284, 95% BootCI [−2.0272; 0.2219]) and body fat (B = 0.5092, 95% BootCI [−0.4756; 1.6934]). These results suggest that high or low values of motivation did not increase or decrease volitional high-intensity exercise, and lower levels of fitness (CRF, muscular and body fat) were associated with higher volitional exercise intensity. These findings highlight the need for regular moderate-to-vigorous physical exercise for maintaining or improving physical fitness, regardless of motivation regulations, and emphasize the importance of new strategies in PEC with acute vigorous-intensity activities that retain the health-enhancing effects.

## 1. Introduction

Despite the numerous benefits of regular physical activity (PA), Western children and adolescents spend too much time in sedentary behaviors, which is worsening every decade [1]. The World Health Organization (WHO) stated that adolescents should achieve at least an average of 60 min per day of moderate-to-vigorous PA (MVPA) and must limit sedentary time [2]. The suggested 150 min per week of moderately vigorous exercise or PA is frequently not achieved by individuals due to a lack of motivation [3].

A school is a place where adolescents spend most of their day. It is known that schools and Physical Education classes (PEC) are privileged spaces and promoters of positive changes for the rest of life [4,5,6], in which time-efficient approach interventions have a prominent role. There are several reasons why PECs should be taught in schools, but the effectiveness of current methods for PEC is frequently challenged, notwithstanding this point of view [7]. It has also been demonstrated that most PEC programs have difficulty attaining the full range of health and educational outcomes included in the PEC curriculum [5]. Professionals constantly point to an overloaded curriculum that creates additional time demands as the main barrier to achieving these parallel educational and health goals [4,6].

Recreational sport and exercise can be performed for their associated enjoyment or for the challenge of participating in an activity [8]. Volition consists of meta-motivational processes; when higher-level action control processes fail, volition may consist of increasing vigor. Research on the importance of volition in the area of exercise psychology is very limited; however, volitional skills have proven to be a sound predictor of performance in other areas of sport, such as elite sports [9].

According to the Self-Determination Theory (SDT), two types of motivation influence personal behavior: the intrinsic (doing a task for the inherent pleasure) and the extrinsic (doing an activity for instrumental reasons, obtaining separable outcomes, or to avoid disapproval). Extrinsically motivated behaviors are expressed in four regulations: external (influenced by external contingencies), introjected (performing to obtain social approval or avoiding internal pressure), identified (recognition and acceptance of the behavior) and integrated (accepting and integrating behavior in other aspects of the self) [10]. The great majority of studies look at the connections between exercise habits and behavioral regulation, and while some have consolidated autonomous and controlled forms of regulation into summary scales or embraced the Relative Autonomy Index (RAI), while the majority have integrated some or all of the particular regulations listed within [8]. RAI represents the self-determination continuum, where lower scores indicate less autonomous motivation and higher scores indicate more autonomous motivation [11].

Intense efforts have been considered inappropriate and not motivating for the general/sedentary population due to feelings of incompetence [12,13]. Notwithstanding the limited number of people willing to engage in moderate-to-vigorous physical exercise (MVPE) and the high attrition of those who participate [14], the evidence shows a high effectiveness of intense exercise to reduce mortality, even considering a long lifespan [15]. According to Bond et al. [16], time spent in high-intensity activities is the most important factor in promoting vascular health and autonomic cardiac modulation. Clearer definitions of the nature of the exercise behaviors under analysis (type, intensity, volume, density), which may differ between studies, as well as their potential interest to the individual, may offer more information on this topic.

The growing evidence of the usefulness of SDT-based interventions for promoting the adoption and maintenance of exercise is a significant advance, but few studies include biological markers of successful exercise-related outcomes [8], such as volitional intensity. Adjusting exercise intensity using heart rate (HR) has been a valid option, mainly in prolonged and submaximal periods. HR has become one of the most used outcomes to assess intensity, and several authors suggest that high-intensity exercise corresponds to a value equal to or higher than 90% HR_max_ [17,18,19].

The present study examined the relationship between physical fitness, motivation and volitional high-intensity exercise in adolescents. This study aimed to assess whether physical fitness had an indirect effect on exercise intensity (through motivation). To the best of our knowledge, no studies formally tested the mediating role of motivation in exercise intensity and physical fitness relationships in PEC, so we hypothesized that motivation (in terms of RAI) exerts a mediating effect on the relationships between cardiorespiratory (CRF) and muscular fitness and exercise intensity in this population.

## 2. Materials and Methods

### 2.1. Trial Design

Based on the need and importance of increasing the knowledge about the relationship between exercise intensity and motivation, we aimed to analyze the level of exercise intensity, motivation and fitness in a sample of a Portuguese adolescent population. Therefore, the main purpose of this study was to analyze the indirect (motivation-mediated) effects of physical fitness on adolescents from the 10th to 12th grades on volitional exercise intensity. More precisely, the study sought to learn how these variables are associated, allowing us to identify possible indicators to guide national strategies in promoting healthy lifestyles in young people. Exploring the mechanisms or processes that mediate the effects of fitness on exercise intensity is crucial for developing effective interventions in PEC.

### 2.2. Participants

Data were retrieved from the baseline assessment and intervention group of a randomized controlled trial investigating the effects of High-Intensity Interval Training (HIIT) in High-School PEC. This project was registered on ClinicalTrials.gov (ID: NCT04022642) and approved by the Ethics Committee of the University of Évora (doc. 19017). In all aspects, this trial was conducted according to the Declaration of Helsinki on Human Research.

Two public schools in the city of Beja (Portugal) were invited to participate. Written consent was obtained from the school principal and parents before the start of the investigation. After an invitation, the researchers met with the school principal and provided information on the project. After accepting to participate, 108 adolescents from the 10th to 12th grades (50 boys and 58 girls, mean age 16.0 ± 0.9 years) and their parents were informed of a detailed description of the scientific background, objectives, and safety. Students were ineligible if they did not provide parental consent to participate, had physical limitations or revealed intellectual disabilities.

### 2.3. Interventions

Throughout the 16 weeks, students took part in the regular 90 min PEC twice a week, conducted by the schools’ PEC teachers following the regular curriculum. Students replaced the warm-ups established in the PEC curriculum with the proposed HIIT training sessions. After the HIIT sessions, students completed the planned PEC.

The HIIT sessions were applied in the first 10–15 min of each PEC, including a brief warm-up, ranged from 14 to 20 all-out bouts intervals adopting a 2:1 work-to-rest ratio and involving a combination of aerobic and body weight muscle-strengthening exercises designed to be fun, engaging, as well as vigorous in nature. Sessions were designed progressively from 4 min in week zero to 10 min in week three using the Tabata protocol (20 s intense work, followed by 10 s rest). From week four to week seven, the same volume of exercises was applied but using 30 s intense work, followed by 15 s rest. From weeks 9 to 15, sessions were completed in pairs (Figure 1).

A cut-point of ≥90% of maximal HR was a criterion for satisfactory compliance to high-intensity exercise. HR has become one of the most used outcomes to assess the intensity, and several authors suggest that each interval corresponds to a value equal to or higher than 90%HR_max_ [17,18,19]. During the supervised intervention, the researchers recorded HR using the Heart Zones Move™ software application, which uses a forearm wearable plethysmography heart rate sensor (Scosche Industries, Oxnard, CA, USA) to ensure compliance with the exercise stimulus at the predetermined target HR zone. In addition, rating perceived exertion (RPE) was also measured in each exercise session to estimate effort, fatigue and training load, targeting >17 on the 6–20 Borg scale [20,21].

### 2.4. Outcomes

Motivation and physical fitness (CRF, upper-body strength) were assessed by the Principal Investigator at the schools participating in the study. Participants’ body composition and body mass assessments were conducted sensitively through the presence of a same-sex research staff when possible. The Principal Investigator provided a brief verbal description and demonstration of each fitness test before evaluation.

#### 2.4.1. Motivation

Motivation was assessed with the Behavioral Regulation in Exercise Questionnaire 3 (BREQ-3) [22]. BREQ-3 is a valid and reliable measurement instrument to measure behavior regulation underlying the self-determination theory in the exercise domain and consists of 18 items with a five-point Likert scale, which varies between 1 (“not true for me”) and 5 (“very true for me”). The scores from each BREQ subscale (amotivation, external, introjected, identified and intrinsic motivation) were weighted and subsequently aggregated to form a solitary numerical index, the RAI, representing the self-determination continuum where lower scores indicate less autonomous motivation, whereas higher scores indicate more autonomous motivation: (amotivation multiplied by −3) + (external regulation multiplied by −2) + (introjected regulation multiplied by −1) + (identified regulation multiplied by 2) + (intrinsic regulation multiplied by 3) [11].

#### 2.4.2. Physical Fitness

CRF was assessed using the Yo-Yo Intermittent Endurance Test level one. This test has been previously confirmed as valid and reliable to assess aerobic fitness and intermittent high-intensity endurance in 9- to 16-year-old children [23]. This test consists of incremental shuttle running starting from the speed of 8 km·h^−1^ until exhaustion. The maximum running speed is 14.5 km·h^−1^. Each shuttle run consists of 2 × 20 m interspersed by 10 s of active recovery (slow jog or walk) for a short 2.5 m shuttle. Within each speed stage, there are several shuttle runs. Running speed is prescribed by a pre-recorded audio track. Participants must reach the 20 m line by the time each audio is heard. The test is finished if the participant is unable to maintain the required speed for the second time during the bout of shuttle running. HR was monitored by telemetric HR during testing. The peak HR recorded during the test was assumed to be representative of maximal HR [24].

Upper-body strength was assessed using the push-up test (FITescola^®^; [25]). The test starts with the participant’s hands and feet touching the floor, and the body in a plank position, with feet apart and the hands positioned below the shoulder line. The participants should lower the body until forming a 90° angle between the arm and the forearm and then return to the starting position. This action was repeated with a previously defined cadence of 20 push-ups per minute.

#### 2.4.3. Body Composition

Participants’ body composition and body mass were measured to the nearest 0.1 kg in light sportswear using bioelectrical impedance analysis (Tanita MC-780, Tokyo, Japan), and height was measured to the nearest 1 mm using a portable stadiometer (Seca 213 Portable Height Measuring Rod Stadiometer, Hamburg, Germany). Body composition measurements were performed through bioelectrical body impedance analysis (BIA). Measurements were performed without accessories that contain metal (earrings, belts, coins), and female adolescents should not have a menstrual period. To ensure normal hydration status for BIA testing, participants were asked to adhere to the following pretest requirements: no vigorous exercise within 12 h of the test and no caffeine or alcohol consumption within 12 h of the test [26]. Both weight and height were measured twice to reduce the risk of measurement error. BMI was calculated using the standard formula (weight [kg]/height [m^2^]).

### 2.5. Sample Size

Power calculations were tested prior to further statistical analysis and based on empirical estimates of sample sizes to detect the Mediated Effect needed for 0.8 Power [27]. For the medium effects (0.39) of both *paths a* and *b*, 78 students were required to detect the mediation effect. *R*^2^ effect-size measures are presented to assess variance accounted for in mediation models.

### 2.6. Data Analysis

All statistical analyses were performed with the Statistical Package for the Social Sciences v.24 (IBM Corp., Armonk, NY, USA). Descriptive statistics were used to characterize the subjects and exercise test results. All variables were assessed for normality using the Kolmogorov–Smirnov test. A bivariate analysis, using the parametric Pearson correlation coefficient or the nonparametric Spearman correlation coefficient (r_s_), was used to indicate the strength of the association between variables. Interpretation of correlation coefficients was as follows: r ≤ 0.49 weak relationship; 0.50 ≤ r ≤ 0.74 moderate relationship; and r ≥ 0.75 strong relationship [28]. All *p*-values were two-tailed, and values below 0.05 were considered to indicate statistical significance.

To examine whether motivation mediated the relationship of physical fitness with exercise intensity, separate models were created for each outcome with physical fitness as a predictor and motivation as a mediator. Figure 2 illustrates the overall mediation models used in the analysis. The diagram on the left side shows the total effect (*path c*), which represents the effect of predictor X on outcome Y without considering mediation. In addition, the total effect of X on Y is equal to the sum of the direct and indirect effects of X. This path quantifies how much two cases that differ by a unit on X are estimated to differ on Y. The diagram on the right side shows a simple mediation model that represents the effect of predictor X on outcome Y including mediation (M). In this model, there are two pathways by which (X) can influence (Y), the indirect effect (*path a.b*) and the direct effect (*path c’*). The indirect effect is the product of *path a* (the effect of the predictor on the mediator) and *path b* (the effect of the mediator on outcomes partially out the effect of the predictor) and represents how Y is influenced by X through M. The indirect effect quantifies how much two cases that differ by one unit on X are estimated to differ by *a.b* units on Y because of the effect of X on M, which, in turn, affects Y. The direct effect represents the effect of (X) on outcome (Y) that cannot be attributed to mediator (M). So, this path quantifies how much two cases that differ by one unit on X are estimated to differ by *c’* units on Y holding M.

Mediation analyses were performed according to Hayes [29], who advises that mediation can occur even when the total effect (the relationship between (X) and (Y), represented by *path c*) is not statistically significant. This is because the total effect is estimated by a different statistical model from the indirect effect, often having a lower power than the indirect effect test. Requiring total effect also ignores the risk of opposing direct and indirect effects, which might combine to produce a nonsignificant total effect. Yet, before testing mediation, *path a* and *path b* were quantified with regression coefficients. The SPSS macro developed by Preacher and Hayes [30] (PROCESS version 4.0) was used to test mediation. This tool tests the significance of indirect effects using bootstrap confidence intervals. The bootstrap method does not require assumptions of normality of the sampling and has higher power and better Type I error control when compared with the product-of-coefficients approach, most well known as the Sobel test [29,30,31]. The indirect effects were deemed significant if 0 was not included in the bootstrap confidence intervals. We report the results of bootstrapping procedures, with the resampling of 10,000 bootstrap replicates (95% confidence intervals, 95% BootCIs). Finally, the indirect effects were described by unstandardized (B) and completely standardized effects (β).

## 3. Results

Data for 108 students (53.7% female) aged 16.0 ± 0.92 years that presented valid volitional intensity data were available for analysis (Table 1). Figure 3A to C show the results of the mediation analysis for each outcome according to the conceptual model (Figure 2). Total effect (*path c*) was significant in models A (CRF) and C (body fat) mediated through motivation. Regarding bivariate correlation, exercise intensity showed positive correlations with body fat (r = 0.318, *p* < 0.01) and a negative correlation with CRF and muscular fitness (ρ = −0.449, *p* < 0.01; r = −0.190, *p* < 0.05).

The bootstrap-derived 95% confidence intervals included zero for all outcomes mediated through motivation, revealing a non-significant, indirect effect of physical fitness through motivation on exercise intensity, specifically on CRF (B = −0.0355, 95% BootCI [−0.5838; 0.4559]; *R*^2^ = 0.17; post hoc power = 1), muscular fitness (B = −0.7284, 95% BootCI [−2.0272; 0.2219]; *R*^2^ = 0.08; post hoc power = 0.86) and body fat (B = 0.5092, 95% BootCI [−0.4756; 1.6934]; *R*^2^ = 0.10; post hoc power = 0.93). There were direct effects (*path c’*) of CRF (*p* < 0.0001) and body fat (*p* < 0.01) on exercise intensity, suggesting that high values of CRF decrease volitional exercise intensity and high values of body fat increase physical exercise intensity, without mediation through motivation.

In *path a* (the effect of physical fitness on motivation), all models reveal a significance but show no significance in *path b* (the effect of motivation on exercise intensity), which is confirmed by bivariate correlations: a negative correlation with body fat (ρ = −0.325, *p* < 0.001) and positive correlations with CRF and muscular fitness (ρ = 0.454, *p* < 0.001; ρ = 0.430, *p* < 0.001). Regarding *path b*, motivation reveals a negative correlation with exercise intensity (ρ = −0.182, *p* = 0.046). Direct effects (*path c’*) remain significant on models A (CRF) and C (body fat) mediated through motivation. Therefore, the absence of an indirect effect suggested that high or low values of RAI did not increase or decrease volitional high-intensity exercise.

## 4. Discussion

The present study examined the relationship between physical fitness, motivation and volitional high-intensity exercise in adolescents. This study aimed to assess whether physical fitness had an indirect effect on exercise intensity (through motivation). The major findings of this study imply that the absence of an indirect effect suggested that high or low values of RAI did not increase or decrease volitional high-intensity exercise. Nevertheless, studies have shown that exercise behavioral regulation was found to be predictive of vigorous exercise, where introjected regulation, identified regulation and intrinsic motivation were positively associated with strenuous exercise behaviors [32]. Still, intrinsic motivation appears to be a more consistent predictor of moderate and vigorous exercise than identified regulation, and autonomous motivation was predictive of long-term moderate-to-vigorous exercise [33]. The mediation mechanism assumes that the independent variable influences the mediator, and the mediator affects the dependent variable. So, the independent variable’s total effect is divided into indirect effects through a mediator. In our work, we used this statistical method to test whether motivation was a mediator of the relationship of physical fitness with volitional exercise intensity. In this study, mediation analyses indicate that adolescents with a higher physical fitness and/or more motivation did not show better biological markers of successful exercise-related outcomes such as volitional intensity. This research adds to the existing literature by examining the indirect effect of physical fitness on volitional exercise intensity in adolescents via motivation. It is critical to define and measure the mechanisms through which physical fitness may be linked to volitional exercise intensity in order to improve recommendations and interventions. This is because investigating mediators can aid in identifying critical aspects that require more attention in order to enhance outcomes.

In children and adolescents, higher amounts of sedentary behavior are associated with increased adiposity and poorer cardiometabolic health and fitness [2]. Lack of leisure time, reduced access to facilities and low motivation to engage in physical activities are frequently reported barriers to strong adherence to exercise programs [34,35]. Intense efforts have been considered inappropriate and not motivating for the general/sedentary population due to feelings of incompetence [12,13]. However, in our study, lower levels of fitness (CRF, muscular and body fat) were associated with higher volitional exercise intensity. Notwithstanding the limited number of people willing to engage in MVPE and the high attrition of those who participate [14], the evidence shows high effectiveness of intense exercise to reduce mortality, even considering a long lifespan [15]. An optimal stimulus that promotes cardiovascular and peripheral adaptations implies several minutes per session in the so-called *red zone*, which usually means a minimum intensity of 90%VO_2max_ [36]. Interventions designed to increase MVPE in PEC indicate that interventions can increase the proportion of time students spend in higher intensities during PEC and reduce sedentary behavior, since motivational climates that emphasize effort and improvement and provide opportunities to demonstrate leadership and make decisions have a positive impact on PA [37]. Physical self-perception could be considered for adherence to MVPE, although Rey et al. [38] suggest that adolescents’ psychological perceptions and health might be improved in response to morphological adaptations, without concomitant improvements in objectively measured physical characteristics or performances.

Despite the novelty and interest of our findings, some limitations must be addressed. First, the use of a cross-sectional design, which prevents the determination of the temporality of the effect of physical fitness and motivation on exercise intensity and the inference of causality from our hypothesized path models, is limited by the use of cross-sectional data. Second, our sample was limited to adolescents from the 10th to 12th grades from a public school in the city of Beja (Portugal). This means that our findings cannot be applied to other populations. Despite this, our selected bootstrapping method has strong statistical power and is considered a useful tool for avoiding Type I errors [29]. Third, other mediator variables may contribute to the links between physical fitness and volitional exercise since most studies opted to set intensity through external load, expressed in speed or distances. Few studies have objectively measured internal load by monitoring HR [34,35,39,40,41,42,43] or RPE [43,44], and only some defined cut lines as high intensity >85%HR_max_ [40,42] or >90%HR_max_ [34]. HR has become one of the most used outcomes to assess intensity. Adjusting exercise intensity using HR is a valid option, mainly in prolonged and submaximal periods. It is expected that HR reaches maximum values (>90–95% HR_max_) close to the speed/power associated with VO_2max_, which does not always happen, especially in very short exercises (<30 s) [17,18]. This may be related to the known delay in HR response at the beginning of exercise, which is slower than the VO_2_ response.

In conclusion, high or low values of RAI did not increase or decrease volitional high-intensity exercise, and lower levels of fitness (CRF, muscular and body fat) were associated with higher volitional exercise intensity. To the best of our knowledge, this is the first study addressing the indirect effect (through motivation) of physical fitness on volitional exercise intensity using a mediation model in a sample of older adolescents; however, future studies are needed to confirm these findings. The idea that public health gains will be higher if we help the least motivated become more motivated due to the lack of sufficient motivation to participate in moderately intense exercise or PA is being challenged. In modern society, it is unlikely that individuals will ever return to the high average PA levels of the past. The results of this study emphasize the importance of new strategies in PEC with acute vigorous-intensity activities. Time-efficient interventions have a preeminent role; moreover, exercise protocols that result in short-term physiological health improvements are of interest to physical education teachers, as well as to rehabilitation, health and exercise professionals.


**Practical Applications**


With this study, the authors aim to provide novel HIIT protocols for schools with less volume (only twice a week) and higher density (less rest in each interval) which include resistance exercises through calisthenic exercises and plyometrics. In addition to the extremely low volume (on average, 10 min/week), it should be noted that there is no need for external loads to implement these protocols; the use of all-out bouts and plyometrics are also simple approaches. The actual PEC time is still restricted due to activities and teaching breaks, as well as absences due to illness, medical appointments, and a lack of appropriate clothing, making it difficult to find content that can positively influence healthy physical fitness in students due to an objective lack of time. Replacing the traditional warm-up without interfering with other curricular content provided in PEC with this time-efficient approach could have a prominent role in improving students’ fitness. These findings highlight the need for regular MVPE for maintaining or improving physical fitness, regardless of motivation regulations, and emphasize the importance of new strategies in PEC with acute vigorous-intensity activities that retain the health-enhancing effects.

## Figures and Tables

**Figure 1 healthcare-11-00800-f001:**
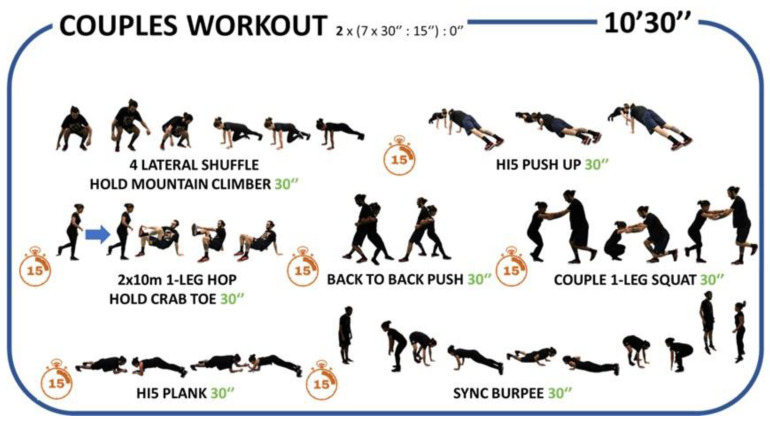
Graphical description of an example session (couples).

**Figure 2 healthcare-11-00800-f002:**
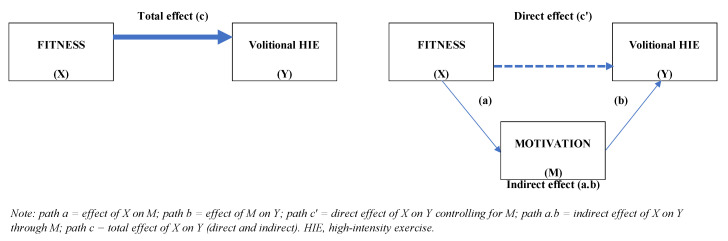
Conceptual model of mediation analysis: indirect effect through motivation.

**Figure 3 healthcare-11-00800-f003:**
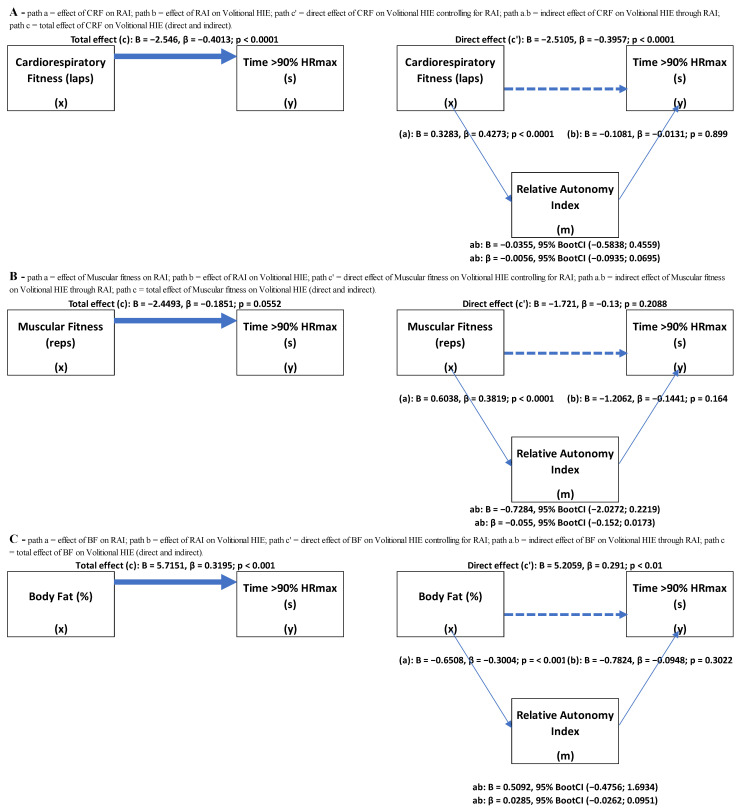
Mediation models showing the effects (total, direct and indirect) of physical fitness variables on volitional high-intensity exercise.

**Table 1 healthcare-11-00800-t001:** Descriptive statistics of study variables (mean ± sd).

	Total
**ANTHROPOMETRIC (n)**	108 (f = 58)
Body Fat (%)	23.9 (7.3)
BMI (kg.m^−2^)	21.6 (3.6)
**MOTIVATION (n)**	108 (f = 58)
Relative Autonomy Index	41.9 (15.5)
**Cardiorespiratory Fitness (n)**	108 (f = 58)
Yo-Yo IE L-1 (laps)	24.0 (18.5)
**Muscular Fitness (n)**	108 (f = 58)
Push-ups (reps)	16.6 (9.8)
**VOLITIONAL HIE (n)**	108 (f = 58)
Average time > 90% HRmax (s)	178.5 (129.9)

BMI: Body mass index; HIE: High-intensity exercise; HR: Heart rate; IE L-1, Intermittent Endurance Test level one.

## Data Availability

The datasets generated during and/or analyzed during the current study are not publicly available due to participants privacy protection but are available from the corresponding author on reasonable request.

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
