# Peer review of "Mediating Effect of Motivation on the Relationship of Fitness with Volitional High-Intensity Exercise in High-School Students"

_healthcare, 2023, doi:10.3390/healthcare11060800_

Round 1

Reviewer 1 Report

1. Lines 15, 26 - do not use abbreviations in the abstract (BC, MVPE), use the full name

2. Lines 63-64 - it is necessary to cite those studies

3. Lines 89-97 - this paragraph should not be here. These are the journal's guidelines for writing an Introduction. Here should be the aim of the study

4. Participants 2.2 - Is it possible that all 108 students did all 16 weeks of the program, without any absences? Or were there more of them in the beginning and 108 entered the final version who implemented the entire program?

5. Intervention Program 2.4. - It is necessary to state how they determined the maximum HR of all participants and, based on that, know what 90% HR is during exercise.

Author Response

RESPONSE TO REVEWER 1

Again, we would like to express our gratitude for the time in reviewing our paper and for providing us comments/suggestions helpful to improve this paper quality.

  • Lines 15, 26 - do not use abbreviations in the abstract (BC, MVPE), use the full name
    • We appreciate this suggestion. Changes have been made to the document

  • Lines 63-64 - it is necessary to cite those studies
    • We thank the reviewer to point this out. We rephased those sentences.

  • Lines 89-97 - this paragraph should not be here. These are the journal's guidelines for writing an Introduction. Here should be the aim of the study
    • Thank for the correction. Changes have been made to the document.

  • Participants 2.2 - Is it possible that all 108 students did all 16 weeks of the program, without any absences? Or were there more of them in the beginning and 108 entered the final version who implemented the entire program?
    • We appreciate this question, thanks for pointing this out. Even though there were more students in the trial, only 108 of them provided reliable intervention data.

  • Intervention Program 2.4. - It is necessary to state how they determined the maximum HR of all participants and, based on that, know what 90% HR is during exercise.
    • Thank again to point this out. We already stated how we determined the HRmax in “2.3.1. Physical Fitness” assessment: “HR was monitored by telemetric HR during testing. The peak HR recorded during the test was assumed to be representative of maximal HR”.

Reviewer 2 Report

Dear authors, first I would like to congratulate you for the manuscript, but I consider that the study should be reconsider after major revisions to be published in Healthcare.

I would like to make some observations that could improve your work always with the authors decision.

The current study aimed to investigate the relationship between physical fitness and motivation in adolescents and analyze if the associations of physical fitness with volitional exercise intensity in adolescents are mediated by motivation. The results of this study reveals the importance of new strategies in Physical Education classes with acute vigorous-intensity activities. Time-efficient interventions have a preeminent role; moreover, exercise protocols that result in short-term physiological health improvements are of interest to physical education teachers, as well as to rehabilitation, health, and exercise professionals.

Abstract:

The information about the intervention program, the HIIT, should be added to the abstract. In line 15, what is “BCs”? In line 22, “CRF” is repeated. In line 26, what is “MVPE”? The first time an abbreviation is written in the article their meaning should be written in extension. 

Introduction:

The introduction offers an adequate contextualization about the empirical literature related to the object of study. However, some imprecisions need correction.

In line 37-39, authors said “The suggested 150 minutes per week of moderately vigorous exercise or PA is frequently not achieved by individuals due to a lack of motivation”. Lack of motivation is the only factor that affects sedentary behaviors?

In line 47-49, the phrase “Professionals consistently cite an overloaded curriculum causing additional time pressures as a primary obstacle to attaining these concurrent educational and health goals in response to the criticism” is confused. Also, the last paragraph of introduction, between lines 89-97, should be removed.

Material and methods:

The methods used in this study are appropriated and well explained to achieve the main objectives of the study. Yet, the hypotheses of the presented study are not clear.

The description of figure 2, between lines 223-237 in confused, authors must simplify that description to facilitate readers comprehension. Authors analyze physical fitness through cardiorespiratory fitness and upper-body strength. Other variables (handgrip strength, p.e.) could also be analyzed.

Results:

The presentation of the results in the textual form is performed in an appropriate way for the reader.

Discussion; Practical applications:

The results presented were interpreted appropriately at the discussion and conclusion sections. These present an adequate and complete interpretation of published literature and discusses its limitations, strengths, and future research. Authors could also have compared the relationship between physical fitness and motivation among sexes and discuss the results with the current literature.

I hope my comments are useful for the authors and can contribute to the improvement of their manuscript.

Author Response

RESPONSE TO REVIEWER 2

We reinforce the gratitude we expressed earlier for the time in reviewing our paper and for providing us comments/suggestions helpful to improve this paper quality.

Abstract:

  • The information about the intervention program, the HIIT, should be added to the abstract. In line 15, what is “BCs”? In line 22, “CRF” is repeated. In line 26, what is “MVPE”? The first time an abbreviation is written in the article their meaning should be written in extension.
    • Thank you for your suggestion. Changes have been made to the document.

Introduction:

The introduction offers an adequate contextualization about the empirical literature related to the object of study. However, some imprecisions need correction.

  • In line 37-39, authors said “The suggested 150 minutes per week of moderately vigorous exercise or PA is frequently not achieved by individuals due to a lack of motivation”. Lack of motivation is the only factor that affects sedentary behaviors?
    • We understand your comment, although as we formulated the sentence, motivation is one of the factor “frequently”, not exclusively.

  • In line 47-49, the phrase “Professionals consistently cite an overloaded curriculum causing additional time pressures as a primary obstacle to attaining these concurrent educational and health goals in response to the criticism” is confused.
    • We agree with your comment. Changes have been made to the document.

  • Also, the last paragraph of introduction, between lines 89-97, should be removed.
    • Thank for the correction. Changes have been made to the document.

Material and methods:

  • The methods used in this study are appropriated and well explained to achieve the main objectives of the study. Yet, the hypotheses of the presented study are not clear.
    • According to this suggestion we have amended the text, adding the hypothesis just before Materials and Methods section.

  • The description of figure 2, between lines 223-237 in confused, authors must simplify that description to facilitate readers comprehension.
    • Thank you for your comment. Considering that this part of the manuscript is critical understanding the mediation analyses applied, and that it is hard to resume, we’ve separated this text in two parts to facilitate readers’ comprehension.
  • Authors analyze physical fitness through cardiorespiratory fitness and upper-body strength. Other variables (handgrip strength, p.e.) could also be analyzed.
    • We appreciate this suggestion, but the only valid physical fitness variables available were CRF and upper-body strength.

Results:

The presentation of the results in the textual form is performed in an appropriate way for the reader.

  • We appreciate this observation.

Discussion; Practical applications:

  • The results presented were interpreted appropriately at the discussion and conclusion sections. These present an adequate and complete interpretation of published literature and discusses its limitations, strengths, and future research. Authors could also have compared the relationship between physical fitness and motivation among sexes and discuss the results with the current literature.
    • Thanks again for the pertinence of the reviewer’s suggestion on genders. Later, we intend to look at these subsamples

Round 2

Reviewer 2 Report

Thanks for the effort by the authors for taking into account the suggestions made.

Best Regards

Author Response

Thanks